# Holistic Inter-Annotator Agreement and Corpus Coherence Estimation in a Large-scale Multilingual Annotation Campaign

**Nicolas Stefanovitch**
European Commission Joint Research Centre
Text and Data Mining Unit
Ispra, Italy
nicolas.stefanovitch@ec.europa.eu

**Jakub Piskorski**
Institute for Computer Science
Polish Academy of Sciences
Warsaw, Poland
jpiskorski@gmail.com

## Abstract

In this paper we report on the complexity of persuasion technique annotation in the context of a large multilingual annotation campaign involving 6 languages and approximately 40 annotators. We highlight the techniques that appear to be difficult for humans to annotate and elaborate on our findings on the causes of this phenomenon. We introduce Holistic IAA, a new word embedding-based annotator agreement metric and we report on various experiments using this metric and its correlation with the traditional Inter Annotator Agreement (IAA) metrics. However, given somewhat limited and loose interaction between annotators, i.e., only a few annotators annotate the same document subsets, we try to devise a way to assess the coherence of the entire dataset and strive to find a good proxy for IAA between annotators tasked to annotate different documents and in different languages, for which classical IAA metrics can not be applied.

## 1 Introduction

In the recent years we have observed an emergence of automated tools for facilitating online media analysis for better understanding of the presented narratives around certain topics across countries, and to identify manipulative, deceptive and propagandistic content. Developing such tools requires annotated data of high quality.

We report on the complexity of annotating such manipulative devices, i.e., persuasion techniques, in the context of a large annotation campaign involving 6 languages and approximately 40 annotators, whose details are described in (Piskorski et al., 2023c). The persuasion technique taxonomy used in the campaign is an extension of the taxonomies used in different shared tasks, contains 23 techniques, and includes i.a., the techniques appealing to emotions, justifications and some forms of logical fallacies. The resulting dataset has been used

in the *SemEval 2023 Task 3: Detecting the Category, the Framing, and the Persuasion Techniques in Online News in a Multi-lingual Setup* (Piskorski et al., 2023b). The primary objective of the work reported in this paper is threefold, namely:

- share some lessons learned from this large multi-lingual annotation campaign that might be beneficial for other researchers planing similar tasks,

- present a detailed analysis of the disagreements between annotators and potential causes thereof and try to measure the complexity of the annotation task, and

- propose a new concept of measuring Inter-Annotator Agreement (IAA) in a multilingual set-up, to overcome the limitations of the classical IAA metrics in such scenario.

We first highlight the techniques that appear to be difficult for humans to annotate using the classical *Cohen's* $\kappa$ (McHugh, 2012), and *Krippendorff's* $\alpha$ (Krippendorff, 2009).

Classical IAA measures impose certain limitations. First, they only capture the coherence of the annotations in texts written in the same language. Secondly, considering annotations done for a single language, there were many annotators, but annotating totally different subsets of documents. The classical IAA metrics are computed using a tiny fraction of the whole dataset: the one where the annotators annotated the same articles, despite the fact that the exact same text could be annotated in different articles by different annotators. Finally, the classical IAA measures only capture agreement at the time of the annotation, but do not tell us anything about the coherence and quality of the final curated dataset.

In order to overcome the aforementioned limitations, we introduce Holistic IAA, a new multilingual word embedding-based IAA metric and

we report on various experiments using it and its correlation with the traditional IAA metrics. However, given somewhat limited and loose interaction between annotators, i.e., only a few annotators annotate the same document subsets, we try to devise a way to assess the coherence of the entire dataset and strive to find a good proxy for IAA between annotators tasked to annotate different documents and in different languages. We present our preliminary results on this research problem with an ultimate goal of establishing a mechanism that allows to compare all annotators no matter which document they annotated, and to detect diverging annotations across languages. Our contributions can be summarized as follows: (i) we measure how *confusing* were the persuasion technique labels for different groups of annotators; (ii) we assess the coherence of the dataset using standard IAA measures; (iii) we introduce a new mutlilingual pan-corpus IAA measure based on semantic similarity; (iv) we exploit this new measure on the raw and curated annotations of the annotators, and compare the resulting ranking of annotators to the one obtained by standard IAA measurements; (v) we comment on the self-coherence of the annotators using the new measure, as well as of the dataset language-wise.

This paper focuses primarily on the annotation agreement and complexity, whereas the description of the resulting dataset is kept to the minimum necessary for understanding the content. For further details please refer to (Piskorski et al., 2023c).

The paper is organized as follows. Section 2 reports on the related work. Section 3 introduces the persuasion technique taxonomy and describes the annotation process. Next, Section 4 reports on the annotation coherence computed using traditional IAA metrics and highlights the hard-to-annotate techniques. Subsequently, Section 5 introduces a new word embedding-based annotator agreement metric and reports on various experiments using it and correlating it with the traditional IAA metrics. We end with some concluding remarks in Section 6.

## 2   Related Work

Persuasion detection in text is related to work on propaganda detection. The work in the latter area initially focused on document-level analysis and predictions, e.g., Rashkin et al. (2017) reports on prediction of 4 classes (*trusted*, *satire*, *hoax*, and *propaganda*) of documents, whereas Barrón-Cedeno et al. (2019) presented a corpus of tagged either as *propaganda* or *non-propaganda*).

In parallel, other efforts focused on the detection of specific persuasion techniques. Habernal et al. (2017, 2018) presented a corpus annotated with 5 fallacies that directly relate to propaganda techniques. A more fine-grained analysis was done by Da San Martino et al. (2019b), who developed a corpus of English news articles labelled with 18 propaganda techniques at span/sentence level. Somewhat related work on detection of use of propaganda techniques in memes is presented in (Dimitrov et al., 2021a), the relationship between propaganda and coordination (Hristakieva et al., 2022), and work studying COVID-19 related propaganda in social media (Nakov et al., 2021a,b). Bonial et al. (2022) reported on the creation of annotated text snippet dataset with logical fallacies for Covid-19 domain. Sourati et al. (2022) presents three-stage evaluation framework of detection, coarse-grained, and fine-grained classification of logical fallacies through adapting existing evaluation datasets, and evaluate various state-of-the-art models using this framework. Jin et al. (2022) proposed the task of logical fallacy detection and a new dataset of logical fallacies found in climate change claims. All the persuasion techniques and logical fallacy taxonomies introduced in the aforementioned works do overlap to a very high degree, but are structured differently, and use different naming conventions.

Various related shared tasks on the detection of persuasion techniques were organized recently, and various taxonomies were introduced (Da San Martino et al., 2020, 2019a; Dimitrov et al., 2021b; Alam et al., 2022; Piskorski et al., 2023b).

Related work on IAA which explores going beyond the limitation of standard measures was reported in (Passonneau and Carpenter, 2014), proposing an idea similar to our in that they are able to compare all annotators between themselves, however, this comparison is done statistically on label distribution while we look at actual content of the annotate textd. Moreover, they are interested in assessing the gold label uncertainty, which is a similar concern to our effort of capturing the label definition difficulty. However, they treat it in a statistical fashion, while we provide simple descriptors. It would be an interesting future work to explore the combination of both approaches.

## 3 Persuasion Technique Annotation

### 3.1 Taxonomy

The taxonomy used in our annotation endeavour is an extension of the taxonomy introduced in Da San Martino et al. (2019b,c). At the top level, there are 6 coarse-grained types of persuasion techniques, namely: *Attack on Reputation*, *Justification*, *Distraction*, *Simplification*, *Call*, and *Manipulative Wording*, whose full definitions are provided in Appendix A. These core types are further subdivided into 23 fine-grained techniques. The 5 new techniques vis-a-vis the taxonomy presented in Da San Martino et al. (2019b,c) are: *Appeal to Hypocrisy*, *Questioning the Reputation*, *Appeal to Values*, *Consequential Oversimplification*, and *Appeal To Time*. The main drive beyond introducing these 5 new techniques is due to their frequent presence in news articles based on our empirical observations. The full two-tier taxonomy, including short definitions, and examples of each fine-grained technique are provided in Figure 3 and 4 in Appendix A respectively.

### 3.2 Annotation Process

Our annotation task consisted of annotating persuasion techniques in a corpus consisting of circa 1600 news articles revolving around various globally discussed topics in six languages: English, French, German, Italian, Polish, and Russian, using the taxonomy introduced earlier. A balanced mix of mainstream media and "alternative" media sources that could potentially spread mis/disinformation were considered for the sake of creating the dataset. Furthermore, sources with different political orientation were covered as well.

The pool of annotators consisted of circa 40 persons, all native or near-native speakers of the language they annotated. Most of the annotators were either media analysts or researchers and experts in (computational) linguistics, where approximately 80% of the annotators had prior experience in performing linguistic annotations of news-like texts. A thorough training was provided to all annotators which consisted of: (a) reading a 60-page annotation guidelines (Piskorski et al., 2023a) — an excerpt thereof is provided in Appendix C), (b) participating in online multi-choice question-like training, (c) carrying out pilot annotations on sample documents, and (d) joint sharing experience with other annotators and discussions with the organisers of the annotation task. Subsequently, each document was annotated by at least two annotators independently. On a weekly basis reports were sent to annotator pairs highlighting complementary and potentially conflicting annotations in order to converge to a common understanding of the task, and regular meetings were held with all annotators to align and to discuss specific annotation cases.

Annotations were curated in two steps. In the first step (*document-level curation*) the independent annotations were jointly discussed by the annotators and a curator, where the latter was a more experienced annotator, whose role was to facilitate making a decision about the final annotations, including: (a) merging the complementary annotations (tagged only by one annotator), and (b) resolving the identified potential label conflicts. In the second step (*corpus-level curation*) a global consistency analysis was carried out. The rationale behind this second step was to identify inconsistencies that are difficult to spot using single-document annotation view and do comparison at corpus level, e.g., comparing whether identical or near-identical text snippets were tagged with the same or a similar label (which should be intuitively the case in most situations). The global consistency analysis sketched above proved to be essential to ensure the high quality of the annotations.

The annotation resulted in annotation of approx. 1600 documents with ca. 37K text spans annotated. The dataset is highly imbalanced. The class distribution and some statistics are provided in Annex B

## 4 Annotation Coherence & Complexity

### 4.1 Traditional IAA

We measured the Inter-Annotator Agreement (IAA) using *Krippendorff's* $\alpha$, achieving a value of 0.342. This is lower than the recommended threshold of 0.667, but we should note that this value represents the agreement level before curation, and as such, it is more representative of the curation difficulty rather than of the quality of the final consolidated annotations. We used the IAA during the campaign to allocate curation roles and to remove low-performing annotators.

We further studied the IAA by ranking the annotators by their performance with respect to the ground truth on the subset of documents they annotated. We split then the annotators into two groups: *top* and *low* based on subjective assessment by the curators after the end of the curation campaign, this assessment was then further confirmed numerically

(see Annex E for details). Their respective average $\alpha$ were 0.415 and 0.250. Finally, we considered the $\alpha$ of the group of the curators, in order to make an approximate estimation of the coherence of the curated dataset, as we expect these curators to consistently curate the data with at least the same coherence they had when annotating documents. There are only two such curators, whose $\alpha$ is of 0.588, which is lower but close to the recommended value.

## 4.2 Confusion matrix

In Figure 1 we present the confusion matrix between the annotations of annotators. A high count denotes both a frequent class and a tendency to confuse the given pair of labels.

One can see that *Loaded Language* (MW:LL) is the single label that is most confused with any other label, and the *Name Calling* (AR:NCL) is the label with which it co-occurs most, and indeed, these two labels have a very similar definition. The same applies to the pair *Casting Doubt* (AR:D) and *Questioning the Reputation* (AR:QCR).

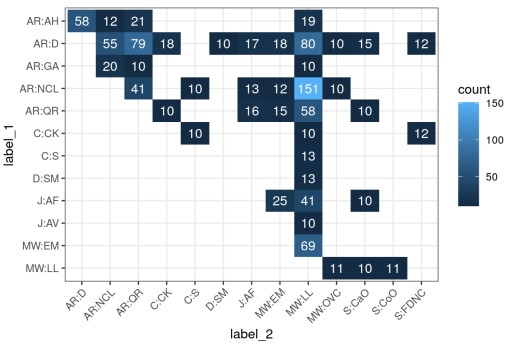

Figure 1: Confusion matrix between single annotations of annotators, thereby denoting tendencies in confusion between the given pairs of labels. Values lower than 10 are blanked.

## 4.3 Techniques' Annotation Complexity

In order to study which persuasion techniques are more difficult to annotate we again divided the annotators in 3 groups: *all* which contains all the annotators, *top* which contains half of the annotators whose performance are the highest as measured by their average *Cohen's* $\kappa$ agreement, and *low* which contains the rest of the annotators.

For each of these groups, and for each of the persuasion techniques, we measured how annotators in a given group tend to disagree with each other – irrespective of the actual ground truth. More precisely, we compute for each pair of annotators and

for all their overlapping annotations the percentage of disagreeing annotations for a given label divided by the total number of annotations between them with that label. Here, annotations of two annotators are considered overlapping if one is at most 10% longer or shorter than the other one, taking into account the exact position of the annotations in the text. We report these numbers in Table 1.

In order to interpret the results, it is also important to take into account that the 2 sub-groups, namely, *top* and *low*, also do interact with each other. We consider the following indicator of complexity: for each of the group if the disagreement is above a given threshold $c$ that we fixed for illustration purpose at 0.25 in the table, the corresponding values are boldfaced. We also divide the techniques in the table (column 'difficulty') into four general annotation complexity classes based on the overall disagreement: very easy ($all \leq .1$, in light green), easy ($all \leq .25$, in green), moderate ($all \leq .4$, in orange), and difficult ($all > .4$, in red).

Additionally, we consider the following indicator: if $top > all$ or if $top > low$ (the techniques for which this applies are marked with an asterisk in the table).

One can see that a high $low$ value does not necessarily mean that the label is actually hard, for instance, the label False Dilemma is very well understood by the $top$ group. High $low$ value and low $top$ value denotes a label whose understanding is not straightforward but does not pose special learning problem, in such case improving annotations for this label requires simply insisting on more basic training.

On the contrary, when the $top$ value is higher than the others (techniques marked with an asterisk), it means that at least one of the groups agrees more with the other group than $top$ group with itself, meaning that there is an inconsistent understanding of the label within the group. This could indicate a difficult label requiring additional clarification to be made to all annotators, or a potential inconsistency in the label definition. This is, for instance, the case for the label Repetition, which is indeed inconsistent as it includes two very different definitions of repetition.

The overall picture of the annotation complexity classes resembles to the per-label performances of classifier systems reported in (Piskorski et al., 2023c), where with a few exceptions the "easiest" labels are the ones with the highest $F_1$ score. It is

important to note that these values are computed on the annotated labels before any curation had taken place, and as such do not reflect the quality of the final dataset, but are more and indication of the intrinsic difficulty of the labels for new annotators.

The class `Doubt` has one of the best reported $F_1$ scores, however, it has a *difficult* annotation complexity, the reason being that it is one of the most confused classes, as it is often a subpart of other techniques.

Some *hard* labels remain a challenge even for `top` annotators, and as such selecting 'reliable' annotators solely based on their overall IAA might not be sufficient to ensure the best quality of annotations, it is also important to identify for which labels additional training might be necessary.

Quantifying the annotation complexity of an annotation campaign in such a way gives an understanding of the difficulty of the task, and allows to identify incoherent understanding of the guidelines early on, and gives a more refined understanding of the quality of the annotations than considering IAA measures alone.

| technique | abrev. | all | top | low | difficulty |
|---|---|---|---|---|---|
| Name Calling-Labeling | AR:NCL | .081 | .096 | **.315** | green * |
| Guilt by Association | AR:GA | .250 | **.393** | **.333** | green * |
| Doubt | AR:D | **.426** | **.286** | **.456** | red |
| Appeal to Hypocrisy | AR:AH | .025 | .033 | .111 | light green * |
| Questioning the Reputation | AR:QR | .266 | .213 | .372 | orange |
| Flag Waving | J:FW | **.286** | **.667** | 1.00 | orange * |
| Appeal to Authority | J:AA | .222 | .100 | 1.00 | green |
| Appeal to Values | J:AV | .190 | .133 | **.667** | green |
| Appeal to Popularity | J:AP | .231 | .200 | 1.00 | green |
| Appeal to Fear-Prejudice | J:AF | .091 | .095 | .158 | light green * |
| Causal Oversimplification | S:CaO | **.368** | .154 | 1.00 | green |
| Consequential Oversimplification | S:CoO | .250 | .182 | .078 | orange * |
| False Dilemma-No Choice | D:FDNC | **.400** | .154 | 1.00 | orange |
| Strawman | D:S | .200 | **.429** | 1.00 | green * |
| Red Herring | D:RH | **.500** | **.600** | 1.00 | red * |
| Whataboutism | D:W | **.500** | **.500** | 1.00 | red |
| Slogans | C:S | .200 | **.333** | .200 | green |
| Conversation Killer | C:CK | .148 | .100 | **.400** | green * |
| Appeal to Time | C:AT | **.333** | **.750** | **.333** | orange * |
| Loaded Language | MW:LL | .042 | .048 | .089 | light green * |
| Obfuscation-Vagueness-Confusion | MW:OVC | **.400** | **.600** | 1.00 | orange * |
| Exaggeration-Minimisation | MW:EM | .208 | .176 | **.429** | green |
| Repetition | MW:R | **.444** | **.667** | **.400** | red * |

Table 1: Per-label disagreement of three groups of annotators: *all*, *top* and *low* and related complexity markers. The colours reflect the respective annotation complexity class. The techniques considered as potentially requiring additional clarification or whose definitions might exhibit inconsistencies are marked with an asterisk.

## 4.4 Disagreement sources

On top of the findings on annotation complexity we additionally summarize here our findings on the sources of disagreements and annotation complexity from the continuous meetings with annotators and curators:

- disregarding small nuances in the definition of *Loaded Language* and *Name Calling* we noticed that disagreements and annotation or non-annotation of some instances were due to subjective perception linked to cultural differences, which was apparent when comparing annotations across languages,

- some annotators had problems with the *Justification* techniques, including, in particular, *Appeal to Popularity*, *Appeal to Values*, *Appeal to Authority* due to not understanding upfront that one subjective opinions on what is considered a value or an authority does not play a role for definition of these techniques, and not considering the role of negation, e.g., not understanding that making a reference to something not being popular falls per definition under *Appeal to Popularity* too,

- many annotators, who probably did not read the guidelines thoroughly, literally interpreted some persuasion technique definitions, e.g., in the context of *Simplification* techniques, instead of detecting certain logic patterns in text (see Annex A for definitions), the annotators literally interpreted the word '*simplification*' and reasoned based on the base of whether the presentation of the information is too simplistic and certain facts were downplayed or exaggerated, which is actually linked to a different technique, i.e., *Exaggeration-Minimisation*,

- some of the media analysts who served as annotators were often using background knowledge (professional bias) to make decisions whether some text fragments are instances of persuasion techniques, which was strictly prohibited by the guidelines; this was mainly related to *Simplifications* and *Distractions*,

- some of the annotators, in particular, media analysts were making a direct link of persuasion technique labeling with fact verification, which was not in line with the guidelines.

To sum up, for the major fraction of persuasion techniques the disagreements resulted not from subjective perceptions of the annotators, but mainly due to not sticking strictly to the definitions provided in the 60-page guidelines and/or professional background bias that lead to misinterpretation of the persuasion technique definitions.

# 5 Embedding-based IAA Assessment

## 5.1 Holistic IAA

We introduce a new measure, namely, *Holistic IAA*, which allows to compare an annotator with any other, even if they did not annotate a single document in common and annotated documents in different languages. This metric exploits the property of multilingual aligned sentence embeddings, which are able to encode with similar vector representations sentences in different language with the same meaning, and different sentences with a similar meaning in a given language.

Formally, we introduce the following holistic agreement between two annotators as $o^{\theta_l,\theta_s}(a_1,a_2)$ where $a_i$ is the function that maps input texts to label for a given annotator $a_i$; and for any two pair of strings $\theta_l$ is the threshold on the length ratio and $\theta_s$ is the threshold on the similarity measure defined for any embedding model $M$ using the cosine distance between the embedding vector of the input strings (we denote it with $o$ for the first letter of the word "holos" in Greek).

We define the set of Comparable Text Pairs (CTP) between two sets of texts $X$ and $Y$ as:

$$CTP_{X,Y}^{\theta_l,\theta_s,M} \quad = \quad \{x,y \in X \times Y : \frac{\min(|x|,|y|)}{\max(|x|,|y|)} > \theta_l,$$
$$sim(M(x),M(y)) > \theta_s\}$$

Using this definition and defining $S(a_i)$ as the function returning all the sentences annotated by annotator $a_i$, we define the Holistic IAA for 2 annotators:

$$o^{\theta_l,\theta_s,M}(a_1,a_2) \quad = \quad \frac{\sum_{x,y \in CTP_{S(a_1),S(a_2)}^{\theta_l,\theta_s}} I_{a_1(x)=a_2(y)}}{|CTP_{S(a_1),S(a_2)}^{\theta_l,\theta_s}|}$$

Extending to groups of annotators $A$ and $B$, we get the more generic formulation:

$$o^{\theta_l,\theta_s,M}(A,B) = \frac{\sum_{a,b \in A \times B} \sum_{x,y \in CTP_{S(a_1),S(a_2)}^{\theta_l,\theta_s}} I_{a_1(x)=a_2(y)}}{\sum_{a,b \in A \times B} |CTP_{S(a),S(b)}^{\theta_l,\theta_s}|}$$

Finally, let $An(D)$ denote the set of annotators of a dataset $D$. We can now define the Holistic IAA value for a dataset as:

$$o_D^{\theta_l,\theta_s,M} \quad = \quad o^{\theta_l,\theta_s,M}(An(D),An(D))$$

In a first step, the embedding for each annotated text span by each annotator is computed and stored in a vector database, and is associated with the following metadata: the document id, the annotator and the label. We use FAISS for the vector database, without quantization and with cosine distance (Johnson et al., 2019). While any multilingual embeddings could be used, we specifically use *LASER embeddings* (Schwenk and Douze, 2017) for simplicity reasons, i.e., our aim is to introduce a new paradigm to IAA computation, and we do not aim at determining which embeddings are the best, which is an exercise of more limited interest given the pace at which new embeddings emerge. Moreover, *LASER* embeddings do not require language specific thresholds. As such, one single cut-off to discriminate similar sentences can be used for all the languages, which is not generally the case for semantic similarity models (Isono, 2022). The drawback of these embeddings is that they are less discriminating than other models as the range of values corresponding to true positives largely intersects with the range of values corresponding to false positives.

In a second step, for each annotator and corresponding annotated text spans, we consider the set of all similar text spans. In Figure 2 we illustrate in detail the behaviour of *LASER* on two sample queries reporting the L2 similarity. For the more complex query Q2, all but one retrieved spans are correct, but a divergence in meaning can be seen with decreasing semantic similarity. We use cosine distance in order to avoid the length of the vector to impact the measure. Moreover, in order to avoid comparing sentences of extremely different sizes, the length of the retrieved span and the query span is constrained by a threshold on the ratio of their respective lengths, i.e., $\theta_l$.

**Q1 "недопустимым" (ru, unacceptable)** : *insupportable* (fr, 0.03, unbearable), *invisibile* (it, 0.03, invisible), *insostenibile* (it, 0.04, unsustainable), *Inacceptable* (fr, 0.05, unacceptable)

**Q2 "tout simplement, un mensonge" (fr, all simply a lie)** : *È tutta una menzogna* (it, 0.04, it is all a lie), *jawne kłamstwo* (pl, 0.06, a clear lie), *questa è una bugia* (it, 0.06, this is a lie), *Énorme mensonge* (fr, 0.07, an enormous lie), *alles wieder eine große Lüge* (de, 0.08, again a big lie), *Wir glauben, dass wir belogen werden* (de, 0.09, we believe we are being lied to), *obficie okłamując* (pl, 0.09, lying profusely), *fatiscenti menzogne* (it, 0.09, crumbling lies), оголтелое враньe (ru, 0.09, rabid lies), *n'en faire qu'une bouchée* (fr, 0.09, deal with it easily), *mensonges éhontés* (fr, 0.09, shameless lies)

Figure 2: Example of a query and retrieved sentences, for each span we report the language, L2 similarity and the English translation.

In Table 2 we provide an example of how text spans and their associated metadata can be queried and retrieved from the database. In this example, the retrieved texts all refer to the same concept as the query text, despite spelling variations in one language and different languages being used. How-

| type | dist. | span | lang. | label | an. |
|---|---|---|---|---|---|
| **query** | - | **religiös-fanatische** | DE | MW:LL | E |
| reply | 0.036 | fanatyków religijnych | PL | AR:NCL | D |

| type | sim. | span | lang. | label | an. |
|---|---|---|---|---|---|
| **query** | - | **fassisti** | IT | MW:LL | A |
| reply | 0.027 | fascista | IT | MW:LL | F |
| reply | 0.031 | faszystą | PL | AR:NCL | D |
| reply | 0.036 | Faschismus | DE | MW:LL | G |
| reply | 0.038 | faszyści | PL | AR:NCL | D |
| reply | 0.038 | фашисты | RU | MW:R | H |
| reply | 0.043 | fascisti | IT | MW:LL | A |

Table 2: Example of a query span and result span fetched from the database, L2 similarity is reported, as well as the associated labels and the annotator identifier.

ever, we can observe that the labels can disagree: This illustrates at the same time the difficulty of distinguishing between *Loaded language* and *Name Calling*, and that some annotators are not consistent in their annotations. Notably, the high rate of confusion between *Loaded language* (MW:LL) and *Name Calling* (AR:NCL) is observable.

### 5.2 Validation: Methodology

In order to validate the approach, we perform rank correlation analysis between the ranking computed by standard IAA techniques and the ones with our approach using *Kendall's Tau* rank correlation coefficient (Kendall, 1938). We consider 2 datasets: the raw annotations of the annotators, and the corpus (dataset of curated documents by curators).

The raw annotations allow us to compute pairwise IAA with *Cohen's* $\kappa$ between annotators, who have annotated the exact same documents. For each annotator, we consider the ranking of the annotators he can be compared to and which have at least 10 annotations in common.

Given the raw annotations dataset, we compute the Holistic IAA $o$ value, and for each annotator we rank all the other annotators to which it can be compared to, as measured by the average level of agreement on labels for semantically similar text spans.

### 5.3 Validation: Results

We compare the ranking of most 'similar' annotators for each annotator computed using *Cohen's* $\kappa$ with the ranking computed using Holistic IAA on the same subset of annotators. We consider 3 rankings: `strict` Cohen's $\kappa$; `same` ranking is done on the same set of documents and annotators as the one used to compute Cohen's $\kappa$; `diff` ranking is done on the same pair of annotators, but strictly on documents that were not jointly annotated by them. We perform a simple grid search over the hyper-

parameters $\theta_s$ and $\theta_l$. In Table 3 we show a sample of the parameters searched, in Annex F we report the results of the full grid search performed. The correlation between `strict` and `same` is on overall higher than when comparing `diff` and `strict` as well as `same` and `diff`, and is even perfect or near perfect for a subset of the parameters. We selected the parameter as $\theta_l = 0$ and $\theta_s = 0.75$ for the rest of the paper, despite these being not the optimal. Optimal parameters are too conservative and as such the CTP set was too small in order to compare all annotators or groups of annotators, and a such prevented from further studying the properties of Holistic IAA.

This proves that the Holistic IAA can be used as a proxy for the pan-document pan-annotators agreement for some specific set of parameters, however, without the possibility to precisely link its value to other standard IAA measures, and with the caveat that the correlation is positive yet not perfect. As such, Holistic IAA can be used mainly to comment on the qualitative difference in agreement between different subsets of annotations.

| ratio | sim | ranking1 | ranking2 | coef. | support |
|---|---|---|---|---|---|
| 0.0 | 0.75 | diff | strict | 0.20 | 10 |
| 0.0 | 0.75 | same | strict | 0.80 | 10 |
| 0.0 | 0.75 | same | diff | 0.26 | 18 |
| 0.50 | 0.90 | diff | strict | -0.33 | 3 |
| 0.50 | 0.90 | same | strict | 0.67 | 10 |
| 0.50 | 0.90 | same | diff | 0.00 | 4 |
| 0.75 | 0.80 | diff | strict | 0.33 | 10 |
| 0.75 | 0.80 | same | strict | 0.93 | 10 |
| 0.75 | 0.80 | same | diff | 0.30 | 13 |

Table 3: Rank correlation between: *Cohen's* $\kappa$ computed on the original data (strict), Holistic IAA computed on the same documents as *Cohen* (same), Holistic IAA computed on all the other documents (diff).

### 5.4 Validation: Error Analysis

We performed an error analysis of the confusions found using Holistic IAA: using the 33k+ confusions found by the approach over the dataset, for each pair of labels we evaluate up to 5 alleged confusions and graded the similarity between the corresponding texts on a 3-tier scale. Two texts are considered: *identical* if the meaning is so close that minor nuance in text would not alter the label chosen (e.g. *"opération spéciale"* (fr) and *"Spezialoperation"* (de) both meaning "special operation"); *close* if the meaning is substantially different, but semantically close enough making the label debatable and worthy to be flagged to a curator for review, for instance one text could be more generic than the other one (e.g. *"finì molto male"* (it) =

*"it ended badly"* and *"durement mise à mal"* (fr) = *"badly impacted"*); *unrelated* if the meaning is unrelated - even if the texts contain the same elements.

A total of 502 data points were annotated. Note that only texts containing at least one space were considered. In Table 4 we report the count in each category, and the *m*ean, *m*in and *m*ax similarity measure as given by the LASER embeddings. When adding the *close* and *identical* classes, we get that in a bit more than half of the cases the approach is able to correctly flag potential incoherent annotations.

We can also see the difficulty of setting cutoff boundaries as the range of minimum and maximum semantic distance is overlapping between all the 3 classes, and with *close* and *identical* having almost the same mean boundaries. We can nevertheless observe that the mean value of *close* is 0.75, making it a reasonable candidate for $\theta_l$.

These results show that about half of the annotations flagged by the system were indeed of interest to the curators. However, as such, the results are highly dependent on the model used. Future work will require to identify embeddings with a larger margin between the classes in order to make the work of the curators more efficient.

| relation | count | mean | std | min | max |
|---|---|---|---|---|---|
| unrelated | 237 | 0.667 | 0.143 | 0.145 | 0.896 |
| close | 142 | 0.751 | 0.058 | 0.643 | 0.970 |
| identical | 123 | 0.846 | 0.091 | 0.661 | 1. |

Table 4: Statistics on the distance for 3 similarity classes using the LASER embeddings and cosine distance on a set of potential confusions flagged by Holistic IAA

### 5.5 Impact of the second curation step

In order to further evidentiate the behavior of Holistsic IAA, we use it to quantify the impact of the corpus-level curation step. This step was performed per-language after the usual document-level curation step was accomplished. The data was sorted per-label and the master curators looked at the overall coherence of the annotated text-span label pairs, the context of the spans was also provided. This step lead to several corrections and is understood to have boosted the overall coherence of the dataset, and should be reflected with a higher $o$ value for the corpus.

In Table 5 we consider the agreement as measured by Holistic IAA after step 1 and 2 of the curation by considering the 4 most active cura-

tors: $a_i$ and $s_i$ denote respectively the agreement percentage between annotators and the support at step $i$. For step 2, the $o$ value is higher, and the average IAA is 1.6 pts higher, while the average intra-annotator agreement (self-agreement) is 3.5 pts higher. This demonstrates that Holistic IAA is able to capture and quantify the positive impact of the corpus-level curation.

In Table 6 we illustrate the impact of excluding *Loaded Language* (MW:LL) and *Name Calling* (AR:NCL) from the dataset as these labels constitute nearly half of the annotations and are frequently confused with each other by annotators in terms of absolute number (but not in proportion) as shown in Figure 1 and Table 1. We observe that the agreement between annotators can be label specific.

In Figure 8 we consider the whole curated dataset and measure the $o$ value between pairs of languages. The columns $a_i$ report the value after step 1 and 2 considering the whole range of labels, while the columns $a'_i$ exclude the two labels MW:LL and AR:NCL. Doing so gives us an understanding of the agreement for the lower populated labels. Please note that all the *Attacks on Reputation* (AR:*) and *Manipulative Wordings* (MW:*) were excluded from the second step of the curation due to time constraints - except for DE and PL. The impact of the second curation step is almost always positive for all pairs of languages, except notably for one language for which the related $o$ values deteriorate and which drags down the intra-language coherence score.

Overall, when considering the corpus we observe a quality increase as measured by the $o$ value.

| cur1 | cur2 | a1 | s1 | a2 | s2 |
|---|---|---|---|---|---|
| A | A | 0.597 | 193778 | 0.603 | 177604 |
| A | B | 0.5 | 57351 | 0.517 | 54503 |
| A | C | 0.586 | 45694 | 0.595 | 183937 |
| A | D | 0.544 | 51327 | 0.548 | 123539 |
| B | B | 0.49 | 10319 | 0.523 | 10210 |
| B | C | 0.451 | 8575 | 0.434 | 29189 |
| B | D | 0.61 | 11688 | 0.625 | 27113 |
| C | C | 0.597 | 3185 | 0.647 | 54566 |
| C | D | 0.471 | 6163 | 0.41 | 62387 |
| D | D | 0.771 | 5473 | 0.798 | 35248 |
| $o$ | | 0.567 | | 0.574 | |
| inter | | $0.483 \pm 0.093$ | | $0.499 \pm 0.101$ | |
| intra | | $0.606 \pm 0.093$ | | $0.641 \pm 0.090$ | |

Table 5: Curated dataset, agreement after step 1 and 2, where $a_i$ and $s_i$ denote respectively the $o$ value between annotators and the support at step $i$.

### 5.6 Multilingual Dataset Coherence Estimation

Knowing the dataset coherence computed using standard IAA measures in a monolingual setting,

| cur1 | cur2 | a1 | s1 | a1' | s1' |
|---|---|---|---|---|---|
| A | A | 0.597 | 193778 | 0.462 | 714 |
| A | B | 0.5 | 57351 | 0.207 | 898 |
| A | C | 0.586 | 45694 | 0.217 | 442 |
| A | D | 0.544 | 51327 | 0.387 | 248 |
| B | B | 0.49 | 10319 | 0.338 | 1426 |
| B | C | 0.451 | 8575 | 0.231 | 562 |
| B | D | 0.61 | 11688 | 0.268 | 314 |
| C | C | 0.597 | 3185 | 0.558 | 154 |
| C | D | 0.471 | 6163 | 0.514 | 140 |
| D | D | 0.771 | 5473 | 0.625 | 64 |
| $o$ | | 0.567 | | 0.323 | |
| inter | | 0.483 ± 0.093 | | 0.304 ± 0.111 | |
| intra | | 0.606 ± 0.093 | | 0.496 ± 0.108 | |

Table 6: Curated dataset: agreement with (left) and without (right) taking into account the two classes `MW:LL` and `AR:NCL`.

and comparing it with values computed using Holistic IAA, we extrapolate from it the coherence of the entire multilingual dataset. Only two curators have jointly annotated the same set of documents while acting as annotators before the curation phase and taking on the curator role, as such we can compute the *Krippendorff's* $\alpha$ between them, which is 0.588, a little under the recommended value. The $o$ value between them on the same data is 0.420. A group of 3 "master curators" covered all the languages and curated most of the dataset. Their average $o$ value on the raw annotations is of 0.565. This higher value illustrates the fact that the coherence of the annotations in the final dataset is higher than when measured on the raw annotations.

We now consider only the curated dataset. In Figure 8 we can observe that the $o$ value intra-language range has an average value of 0.538, slightly above the $o$ value of 0.420 of the two reference annotators for which *Krippendorff's* $\alpha$ could be computed. We can conclude that the coherence of the dataset restricted to each language is above the coherence of the reference annotators.

However, most of the inter-language $o$ values are much lower than the intra-language values. We believe this to be due to 2 factors: 1) each curation was performed per-language, ignoring the others, thereby increasing the self coherence of each language; 2) as in the case of the `diff` vs. `strict` in Figure 3 Holistic IAA is less able to capture agreement than in the case of `same` vs. `strict`, thereby denoting a limitation of our approach. This could be partially alleviated by using 'better' embeddings. Nevertheless, even with a lower performance, a tool based on Holistic IAA to check for annotation coherence across languages would help to increase the quality of the dataset by flagging potential inconsistent annotations.

In Table 7 we can observe that the $o$ value for

the dataset is consistently higher after the second curation step vis-a-vis after the first step, suggesting that this new curation approach is of interest to increase the quality of annotations.

| $\theta_s$ | 0.75 | 0.80 | 0.85 | 0.90 | 0.95 |
|---|---|---|---|---|---|
| step 1 | 0.537 | 0.580 | 0.729 | 0.738 | 0.828 |
| step 2 | 0.560 | 0.592 | 0.756 | 0.762 | 0.851 |

Table 7: $o$ value for the curated data after each curation step for different values of $\theta_s$ and fixed $\theta_l = 0$

| lang1 | lang2 | a1 | a2 | a1' | a2' | change | change' |
|---|---|---|---|---|---|---|---|
| FR | FR | 0.612 | 0.653 | 0.45 | 0.47 | +0.041 | +0.02 |
| FR | IT | 0.585 | 0.592 | 0.168 | 0.223 | +0.007 | +0.055 |
| FR | PL | 0.435 | 0.401 | 0.451 | 0.484 | -0.034 | +0.033 |
| FR | RU | 0.43 | 0.443 | 0.245 | 0.288 | +0.013 | +0.043 |
| IT | IT | 0.593 | 0.6 | 0.458 | 0.458 | +0.007 | 0.0 |
| IT | PL | 0.545 | 0.549 | 0.384 | 0.346 | +0.004 | -0.038 |
| IT | RU | 0.509 | 0.524 | 0.229 | 0.234 | +0.015 | +0.005 |
| PL | PL | 0.771 | 0.798 | 0.625 | 0.476 | +0.027 | -0.149 |
| PL | RU | 0.608 | 0.618 | 0.312 | 0.255 | +0.01 | -0.057 |
| RU | RU | 0.501 | 0.531 | 0.343 | 0.373 | +0.03 | +0.03 |
| $o$ | | 0.567 | 0.574 | 0.323 | 0.346 | +0.013 | +0.023 |
| inter | | 0.519 | 0.521 | 0.298 | 0.305 | +0.002 | +0.007 |
| intra | | 0.619 | 0.646 | 0.469 | 0.444 | +0.027 | -0.025 |

Table 8: Curated dataset: impact of the two curation steps on the overall agreement between languages (inter) and inside languages (intra), when the labels `MW:LL` and `AR:NCL` are included (a1 and a2) and excluded (a1' and a2')

# 6 Conclusions

We reported on the complexity of annotating persuasion techniques in a large-scale multilingual annotation campaign. We introduced the Holistic IAA paradigm, a new measure to serve as a proxy of the estimation of inter-annotator agreement and actual corpus coherence in settings that are fundamentally outside the scope of usual IAA measures. We demonstrate that annotator ranking computed using this new measure is positive and can highly correlates with ranking computed using Cohen's Kappa in some settings. Using it, we can observe the beneficial impact of the second step of our 2-step curation phase, and also identify similarity and divergence between annotators for some subsets of labels. The experiment conducted in this study supports what was informally remarked regarding the estimation of the performance of the annotators and increased our confidence in the coherence of the final corpus. We believe that using Holistic IAA as part of the monitoring of multilingual or monolingual large-scale annotation campaigns could help to spot problems by flagging potential incoherence in the labels of semantically similar sentences at an early stage. In future work we envisage exploration of thresholds for finer interpretation and exploring the use of other semantic similarity models.

## Limitations

**Distribution Representativeness** Although the underlying corpus of annotated news articles covers a wide range of topics as well as media from all sides of the political spectrum it should neither be seen as representative nor balanced in any specific way w.r.t. media in any country. Consequently, the distribution of the annotated persuasion techniques might, in principle, not be fully representative as well.

**Biases** Given that human data annotation involves some degree of subjectivity we created a comprehensive 60-page annotation guidelines document to clarify important cases during the annotation process. Nevertheless, some degree of intrinsic subjectivity might have impacted the techniques picked up by the annotators during the annotation, and impacted so the distribution thereof in the final dataset. Furthermore, although the taxonomy used in this annotation campaign covers most of the 'popular' techniques used in the media, we identified some persuasive attempts which could not have been matched with any of the techniques in the existing taxonomy, and were tagged as OTHER (less than 3% of all annotations) and were not considered in the reported work, which once again poses a certain limitation with respect to the representativeness of persuasion technique types used in the media.

**Methodology Soundness** Our results are limited to certain extent, in particular, the introduced IAA metric should be considered as a proof of concept since certain approximations and simplifications were made and parameters were chosen, e.g., the choice for cutoff of maximal retrieved similar sentences, the length ratio to select sentence to be compared is constrained, and the choice of similarity metrics for computing semantic similarity that exploits a specific sentence embeddings model. Different settings and choices could yield different results. Disregarding of these shortcomings, the new metric helped to circumvent the limited scope and utility of classical IAA in such a large-scale multilingual campaign. We believe that the proposed methodology presented in this paper is too some extent generic, and would be of great interest to the community.

The approach considers only the text of the annotation, as such their context is ignored. This limitation is mitigated in case the annotation guidelines do not specify that the span of annotation must contain all necessary information to unambiguously determine the label, which is the case in the campaign whose data was used to illustrate our approach.

## Ethics Statement

**Biases** The news articles for the creation of the underlying dataset were sampled in such a way in order to have a balanced representation with respect to different points of view and type of media. We also strived to engage a mix of annotators with different backgrounds, i.e., both media analysts and computational linguists. Furthermore, the annotators were explicitly instructed not take their personal feeling about the particular topic and to objectively focus on identifying whether specific persuasion techniques were used. Disregarding the aforementioned efforts, the distribution of the various persuasion techniques annotated might not perfectly reflect the broader spectrum of the media landscape in the target languages, which should be taken into account in exploiting the related statistical information for any kind of analysis, etc. Analogously, the findings and statistics related to the annotation complexity are linked to the specific pool of annotators engaged in the campaign, and, consequently, they should be considered as approximative.

**Intended Use and Misuse Potential** The reported work focuses solely on sharing experience with the research community on annotating persuasion techniques in news articles in a large campaign, analysis of the difficulty of annotating such techniques, and ways of measuring annotation agreement and consistency across languages. The reported work is not linked to a release of the underlying annotated dataset, which is a subject of different publication and related ethical considerations.

## Acknowledgements

We are greatly indebted to all the annotators from different organizations, who participated in the annotation campaign, and without whom carrying out the reported study would not have been possible.

The work presented in this paper is financed by the Joint Research Centre of the European Commission.

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

## A  Persuasion Techniques

The two-tier persuasion technique taxonomy has 6 coarse-grained categories:

**Attack on reputation:** The argument does not address the topic, but rather targets the participant (personality, experience, deeds) in order to question and/or to undermine their credibility. The object of the argumentation can also refer to a group of individuals, an organization, an object, or an activity.

**Justification:** The argument is made of two parts, a statement and an explanation or an appeal, where the latter is used to justify and/or to support the statement.

**Simplification:** The argument excessively simplifies a problem, usually regarding the cause, the consequence or the existence of choices.

**Distraction:** The argument takes focus away from the main topic or argument to distract the reader.

**Call:** The text is not an argument, but an encouragement to act or to think in a particular way.

**Manipulative wording:** the text is not an argument, but uses specific language, which contains words or phrases that are either non-neutral, confusing, exaggerating, loaded, etc., in order to impact the reader emotionally.

They are further subdivided into 23 fine-grained persuasion techniques. The full list of the fine-grained techniques is presented in 3, whereas some examples of text snippets representing various persuasion techniques are provided in Figure 4.

## B  Dataset Statistics

In Figure 5 we provide the distribution of the persuasion techniques per language. *Name Calling* and *Loaded Language* are by far the most populated classes across all languages, and are followed by *Doubt* and *Questioning the Reputation*. In total there were approx. 9K text spans (with persuasion techniques) annotated for English (536 documents), 7.2K for French (211 documents), 5.7K for German (177 documents), 8K for Italian (303 documents), 3.8K for Polish (194 documents), and 4.1K for Russian (191 documents).

## C  Annotation guidelines excerpt

This section provides an excerpt from the annotation guidelines (Piskorski et al., 2023a). The following general rules should be applied when annotating persuasion techniques:

- if one has doubts whether a given text fragment contains a persuasion technique then do not annotate it, (*conservative approach*)

- select the minimal amount of text[1] to annotate in case of doubts whether to include a longer text fragment or not,

---

[1]In our guidelines we do have specific rules for each of the persuasion techniques of what the annotation should include, e.g., in case of *Justification* techniques the annotation should include certain appeal and the claim or idea it supports, if explicitly expressed in the immediate context, or, in case of *Loaded Language* only the emotionally-loaded word/phrase should be annotated, disregarding the context it appears in

**ATTACK ON REPUTATION**

**Name Calling or Labelling [AR:NCL]:** a form of argument in which loaded labels are directed at an individual, group, object or activity, typically in an insulting or demeaning way, but also using labels the target audience finds desirable.

**Guilt by Association [AR:GA]:** attacking the opponent or an activity by associating it with a another group, activity or concept that has sharp negative connotations for the target audience.

**Casting Doubt [AR:D]:** questioning the character or personal attributes of someone or something in order to question their general credibility or quality.

**Appeal to Hypocrisy [AR:AH]:** the target of the technique is attacked on its reputation by charging them with hypocrisy/inconsistency.

**Questioning the Reputation [AR:QR]:** the target is attacked by making strong negative claims about it, focusing specially on undermining its character and moral stature rather than relying on an argument about the topic.

**JUSTIFICATION**

**Flag Waiving [J:FW]:** justifying an idea by exhaling the pride of a group or highlighting the benefits for that specific group.

**Appeal to Authority [J:AA]:** a weight is given to an argument, an idea or information by simply stating that a particular entity considered as an authority is the source of the information.

**Appeal to Popularity [J:AP]:** a weight is given to an argument or idea by justifying it on the basis that allegedly "everybody" (or the large majority) agrees with it or "nobody" disagrees with it.

**Appeal to Values [J:AV]:** a weight is given to an idea by linking it to values seen by the target audience as positive.

**Appeal to Fear, Prejudice [J:AF]:** promotes or rejects an idea through the repulsion or fear of the audience towards this idea.

**DISTRACTION**

**Strawman [D:SM]:** consists in making an impression of refuting an argument of the opponent's proposition, whereas the real subject of the argument was not addressed or refuted, but instead replaced with a false one.

**Red Herring [D:RH]:** consists in diverting the attention of the audience from the main topic being discussed, by introducing another topic, which is irrelevant.

**Whataboutism [D:W]:** a technique that attempts to discredit an opponent's position by charging them with hypocrisy without directly disproving their argument.

**SIMPLIFICATION**

**Causal Oversimplification [S:CaO]:** assuming a single cause or reason when there are actually multiple causes for an issue.

**False Dilemma or No Choice [S:FDNC]:** a logical fallacy that presents only two options or sides when there are many options or sides. In extreme, the author tells the audience exactly what actions to take, eliminating any other possible choices.

**Consequential Oversimplification [S:CoO]:** is an assertion one is making of some "first" event/action leading to a domino-like chain of events that have some significant negative (positive) effects and consequences that appear to be ludicrous or unwarranted or with each step in the chain more and more improbable.

**CALL**

**Slogans [C:S]:** a brief and striking phrase, often acting like emotional appeals, that may include labeling and stereotyping.

**Conversation Killer [A:CK]:** words or phrases that discourage critical thought and meaningful discussion about a given topic.

**Appeal to Time [C:AT]:** the argument is centred around the idea that time has come for a particular action.

**MANIPULATIVE WORDING**

**Loaded Language [MW:LL]:** use of specific words and phrases with strong emotional implications (either positive or negative) to influence and convince the audience that an argument is valid.

**Obfuscation, Intentional Vagueness, Confusion [MW:OVC]:** use of words that are deliberately not clear, vague or ambiguous so that the audience may have its own interpretations.

**Exaggeration or Minimisation [MW:EM]:** consists of either representing something in an excessive manner or making something seem less important or smaller than it really is.

**Repetition [MW:R]:** the speaker uses the same phrase repeatedly with the hopes that the repetition will lead to persuade the audience.

Figure 3: Persuasion techniques taxonomy. The six coarse-grained techniques are subdivided into 23 fine-grained ones. An acronym for each technique is given in squared brackets.

**Name Calling or Labelling:** *'Fascist' Anti-Vax Riot Sparks COVID Outbreak in Australia.*

**Guilt by Association:** *Manohar is a big supporter for equal pay for equal work. This is the same policy that all those extreme feminist groups support. Extremists like Manohar should not be taken seriously.*

**Casting Doubt:** *This task is quite complex. Is his professional background, experience and the time left sufficient to accomplish the task at hand?*

**Appeal to Hypocrisy:** *How can you demand that I eat less meat to reduce my carbon footprint if you yourself drive a big SUV and fly for holidays to Bali?*

**Questioning the Reputation:** *I hope I presented my argument clearly. Now, my opponent will attempt to refute my argument by his own fallacious, incoherent, illogical version of history*

**Flag Waiving:** *We should make America great again, and restrict the immigration laws.*

**Appeal to Authority:** *Since the Pope said that this aspect of the doctrine is true we should add it to the creed.*

**Appeal to Popularity:** *Because everyone else goes away to college, it must be the right thing to do.*

**Appeal to Values:** *It's standard practice to pay men more than women so we'll continue adhering to the same standards this company has always followed.*

**Appeal to Fear, Prejudice:** *It is a great disservice to the Church to maintain the pretense that there is nothing problematic about Amoris laetitia. A moral catastrophe is self-evidently underway and it is not possible honestly to deny its cause.*

**Strawman:** *Referring to your claim that providing medicare for all citizens would be costly and a danger to the free market, I infer that you don't care if people die from not having healthcare, so we are not going to support your endeavour.*

**Red Herring:** *Lately, there has been a lot of criticism regarding the quality of our product. We've decided to have a new sale in response, so you can buy more at a lower cost!.*

**Whataboutism:** *A nation deflects criticism of its recent human rights violations by pointing to the history of slavery in the United States.*

**Causal Oversimplification:** *School violence has gone up and academic performance has gone down since video games featuring violence were introduced. Therefore, video games with violence should be banned, resulting in school improvement.*

**False Dilemma or No Choice:** *There is no alternative to Pfizer Covid-19 vaccine. Either one takes it or one dies.*

**Consequential Oversimplification:** *If we begin to restrict freedom of speech, this will encourage the government to infringe upon other fundamental rights, and eventually this will result in a totalitarian state where citizens have little to no control of their lives and decisions they make*

**Slogans:** *"Immigrants welcome, racist not!"*

**Conversation Killer:** *I'm not so naive or simplistic to believe we can eliminate wars. You can't change human nature.*

**Appeal to Time:** *This is no time to engage in the luxury of cooling off or to take the tranquilizing drug of gradualism. Now is the time to make real the promises of democracy. Now is the time to rise from the dark and desolate valley of segregation to the sunlit path of racial justice.*

**Loaded Language:** *They keep feeding these people with trash. They should stop.*

**Obfuscation, Intentional Vagueness, Confusion:** *Feathers can not be dark, because all feathers are light!*

**Exaggeration or Minimisation:** *From the seminaries, to the clergy, to the bishops, to the cardinals, homosexuals are present at all levels, by the thousand*

**Repetition:** *Hurtlocker deserves an Oscar. Other films have potential, but they do not deserve an Oscar like Hurtlocker does. The other movies may deserve an honorable mention but Hurtlocker deserves the Oscar.*

Figure 4: Examples of text snippets with persuasion techniques. The text fragments highlighted in bold are the actual text spans annotated.

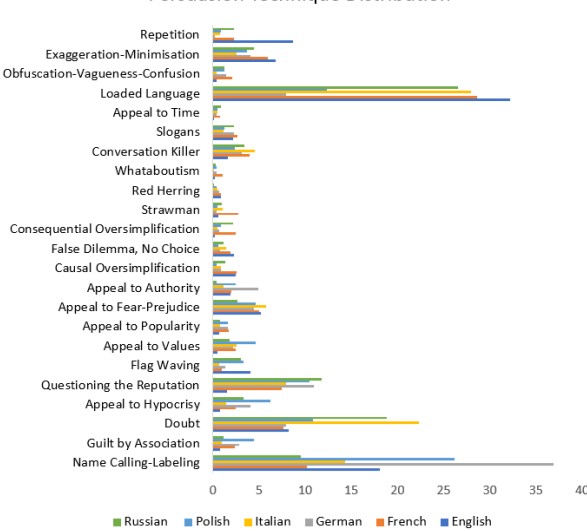

Figure 5: Distribution of the persuasion techniques per language (in percentage).

- avoid personal bias (i.e., opinion and emotions) on the topic being discussed as this has nothing to do with the annotation of persuasion techniques,

- do not exploit external knowledge to decide whether given text fragment should be tagged as a persuasion technique,

- do not confuse *persuasion technique detection* with *fact checking*. A given text fragment might contain a claim which is known to be true, but that does not imply there are no persuasion techniques to annotate in this particular text fragment,

- often, authors use *irony* (not being explicitly part of the taxonomy), which in most cases serves a purpose to persuade the reader, most frequently to attack the reputation of someone or something. In such cases the respective persuasion technique type should be used, or *other* if the use of irony does not fall under any persuasion technique type in the taxonomy,

- in case of quotations or reporting of what a given person said the annotation of the persuasion techniques within the boundaries of that quotation should be done from the perspective of that person who is making some statement or claim (*point of reference*) and not from the author perspective.

For each persuasion technique we have also specified what text fragment should be annotated in the document. The general rule is to annotate the minimum amount of text that can be considered as a trigger to spot the technique, even if it requires an understanding of the context that spans over more than one of the preceding sentences. Sometimes, the to-be-annotated text fragment might go beyond the boundaries of one single sentence. In the following we briefly summarize the rules for all the techniques.

**Name Calling or Labelling:** The noun phrase, the adjective that constitutes the label and/or the name. If quotation marks are used, they should be included in the annotation as well.

**Guilt by Association:** The part of text that refers to an entity and a mention of someone else (considered evil/negative) doing the same or similar thing that is considered negative. The mention of the activity of the target entity might be implicit.

**Casting Doubt:** Only the text fragment that questions the credibility and the object whose credibility is being questioned. There is no need to include the full context.

**Appeal to Hypocrisy:** The text phrase embracing a certain activity, and another one which is used as an argument to accuse the former as being a hypocrite.

**Questioning the Reputation:** Only the text fragments that refer to something negative being mentioned about the person/group/object.

**Flag Waving:** The part of the text that refers to patriotism or other group related values, and the conclusion/action it is supposed to support if it is present in the text.

**Appeal to Authority:** The part of the text that refers to the authority (and potentially some of his/her statement/opinion/action), and the conclusion it supports, in case the latter is present in the text.

**Appeal to Popularity:** The part of the text that refers to something that a majority does or seems to be widely supported and/or is popular together with the conclusion it is supposed to support.

**Appeal to Values:** The part of the text that refers to values, and include the conclusion it is supposed to support, in case the latter is included explicitly in the text.

**Appeal to Fear, Prejudice:** The part of the text that

refers to the fears, prejudices, e.g., of something that might happen.

**Strawman:** When this technique is used, usually the relevant context might span across more sentences. However, one should only annotate the text fragment (sentence or part thereof), which introduces the distraction.

**Red Herring:** When this technique is used, usually the relevant context might span across more sentences. However, one should only annotate the text fragment (sentence or part thereof), which introduces the distraction.

**Whataboutism:** When this technique is used, usually the relevant context might span across multiple sentences. However, one should only annotate the text fragment (sentence or part thereof) that introduces the distraction.

**Causal Oversimplification:** The minimal text fragment that matches the logical pattern should be annotated:

```
Y occurred after X; therefore,
X was the only cause of Y

X caused Y; therefore, X was the only cause of Y
(although A,B,C...etc. also contributed to Y.)
```

or a false conclusion drawn therefrom should be annotated, although, often not all parts of the pattern above are explicitly mentioned in the text.

**False Dilemma or No Choice:** The minimal text fragment that matches one of the following logical patterns should be annotated:

```
(a) Black & White Fallacy:

There are only two alternatives A and B to a given
problem/task. It cannot be A. Therefore, the only
solution is B (since A is not an option).

(b) Dictatorship

The only solution to a given problem/task is A.
```

although, often not all parts of the pattern above are explicitly mentioned in the text.

**Consequential Oversimplification:** The entire text fragment that matches the above logical pattern should be annotated:

```
if A will happen then B, C, D, ... will happen

where:
- A is something one is trying to reject (support)
- B, C, D are perceived as some potential
negative (positive) consequences happening if A
happens.
```

**Slogans:** The slogan only (no need to annotate the conclusion it supports), and in case it is surrounded

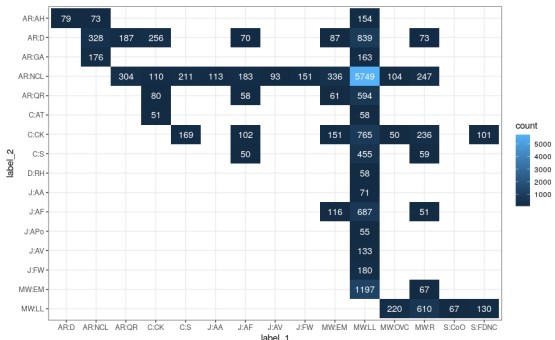

Figure 6: Confusion matrix on the annotations, as per Holistic IAA, for a minimal support of 50

by quotation marks, include them as well.

**Conversation Killer:** A minimal text span that triggers ending the conversation, discussion, etc.

**Appeal to Time:** A minimal text span referring to the argument of time that calls for some action. Both the call and the action should be annotated.

**Loaded Language:** Only the phrase containing loaded words, the context in which they appear should not be annotated. As a general rule one should consider to tag longer text fragment if and only if each of the words adds more emotional 'load' to the text fragment.

**Obfuscation, Intentional Vagueness, Confusion:** The minimal text fragment that introduces confusion: it could be a word, but also a longer piece of text that requires to be read in order to understand the confusion it causes.

**Exaggeration or Minimisation:** The text fragment that provides the description that downplays or exaggerates the object of criticism. The latter should be included in the annotated text as well.

**Repetition:** All text fragments that repeat the same message or information that was introduced earlier. The first occurrence of the message/information is to be annotated as well. If it is not clear what exactly to annotate then the entire sentence should be annotated. Furthermore, it is important to emphasize that a repetition of something per se is not always a persuasion technique, but could sometimes be used only to refer to a topic/issue being discussed.

## D Confusion Matrix based on Holistic IAA

In Figure 7 we report the confusion matrix found using Holistic IAA on the final curated dataset. De-

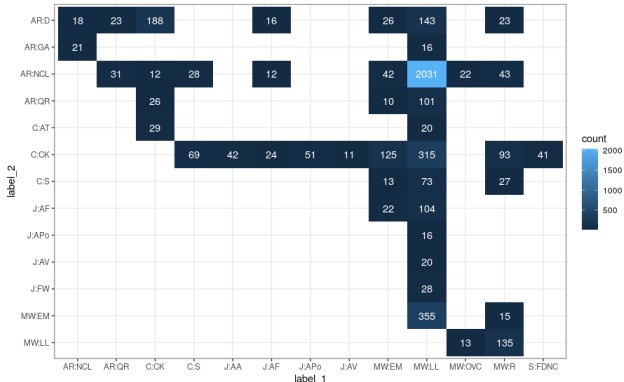

Figure 7: Confusion matrix on final corpus, as per Holistic IAA, for a minimal support of 10

| ratio | sim | ranking1 | ranking2 | coef. | support |
|---|---|---|---|---|---|
| 0.0 | 0.75 | diff | strict | 0.20 | 10 |
| 0.0 | 0.75 | same | strict | 0.80 | 10 |
| 0.0 | 0.75 | same | diff | 0.26 | 18 |
| 0.0 | 0.80 | diff | strict | 0.27 | 10 |
| 0.0 | 0.80 | same | strict | 0.87 | 10 |
| 0.0 | 0.80 | same | diff | 0.20 | 14 |
| 0.0 | 0.85 | diff | strict | 0.26 | 9 |
| 0.0 | 0.85 | same | strict | 0.93 | 10 |
| 0.0 | 0.85 | same | diff | 0.51 | 11 |
| 0.0 | 0.90 | diff | strict | -0.33 | 3 |
| 0.0 | 0.90 | same | strict | 0.67 | 10 |
| 0.0 | 0.90 | same | diff | 0.17 | 4 |
| 0.50 | 0.75 | diff | strict | 0.27 | 10 |
| 0.50 | 0.75 | same | strict | 0.87 | 10 |
| 0.50 | 0.75 | same | diff | 0.26 | 18 |
| 0.50 | 0.80 | diff | strict | 0.40 | 10 |
| 0.50 | 0.80 | same | strict | 0.87 | 10 |
| 0.50 | 0.80 | same | diff | 0.20 | 14 |
| 0.50 | 0.85 | diff | strict | 0.33 | 9 |
| 0.50 | 0.85 | same | strict | 0.93 | 10 |
| 0.50 | 0.85 | same | diff | 0.57 | 11 |
| 0.50 | 0.90 | diff | strict | -0.33 | 3 |
| 0.50 | 0.90 | same | strict | 0.67 | 10 |
| 0.50 | 0.90 | same | diff | 0.00 | 4 |
| 0.75 | 0.75 | diff | strict | 0.27 | 10 |
| 0.75 | 0.75 | same | strict | 0.87 | 10 |
| 0.75 | 0.75 | same | diff | 0.46 | 14 |
| 0.75 | 0.80 | diff | strict | 0.33 | 10 |
| 0.75 | 0.80 | same | strict | 0.93 | 10 |
| 0.75 | 0.80 | same | diff | 0.30 | 13 |
| 0.75 | 0.85 | diff | strict | 0.18 | 9 |
| 0.75 | 0.85 | same | strict | 0.67 | 10 |
| 0.75 | 0.85 | same | diff | 0.22 | 11 |
| 0.75 | 0.90 | diff | strict | -0.33 | 3 |
| 0.75 | 0.90 | same | strict | 0.67 | 10 |
| 0.75 | 0.90 | same | diff | 0.00 | 4 |

Table 9: Rank correlation between: *Cohen's* $\kappa$ computed on the original data (strict), Holistic IAA computed on the same documents as *Cohen* (same), Holistic IAA computed on all the other documents (diff).

spite overall similarity with exact confusion matrix on annotations in Figure 1 there are a few notable difference, particularly with C:CK being largely largely more confused and AR:D being largely less confused.

In Figure 6 we report the confusion matrix on the set of annotations. It is closer the Figure 1 than Figure 7. This indicates that the curation process actually eliminated some common confusion in the annotations. The magnitude are different for two reasons that can not be measure independently: it contains less errors, and there are less overall pair-wise comparison performed as the total set of annotation considered is about two times smaller.

# E Identifying the *top* and *low* groups of annotators

In order to split the annotators into two groups, in a first time the curators based on their subjective assessments established 2 groups of equal size. This was further corroborated in a second step using the following approach: the curated data was taken as ground truth and the annotators were considered as classifiers, whose annotations are considered as prediction. As such we computed the micro $F_1$ for each annotator, and ranking them along that measure, the median split validated the first subjective assessment which contained a few more annotators, which all ranked the highest in the lower split. The average micro $F_1$ score of the top and low groups are respectively of 0.603 +- 0.119 and 0.284 +- 0.081.

# F Parameter search

We conducted an exhaustive parameter search in order to determine the optimal parameter $\theta_l$ and $\theta_s$

which maximise the rank correlation between the ranking of annotators produces by Cohen's Kappa and the one produces by Holistic IAA. We consider 3 groups of document-annotators pairs: strict, for which Cohen's $\kappa$ can be computed; same any document annotated by previous annotators; diff any documents not annotated jointly by previous annotators. The pairwise comparison of these sets with *Kendall's Tau* rank correlation is consider as 3 dimensions of a multi-criteria decision problem: A minimal number of 10 annotations in common is required for a pair of annotator to be considered, support is the total number of pairs being compared. The table with the result of the parameter search are reported in Table 9.