# OpenReview forum: "Holistic Inter-Annotator Agreement and Corpus Coherence Estimation in a Large-scale Multilingual Annotation Campaign"
_EMNLP/2023/Conference — EMNLP 2023 Main_

### Official Review · Reviewer_Jcj8 · 2023-08-03

**Soundness:** 4

**Excitement:**

3: Ambivalent: It has merits (e.g., it reports state-of-the-art results, the idea is nice), but there are key weaknesses (e.g., it describes incremental work), and it can significantly benefit from another round of revision. However, I won't object to accepting it if my co-reviewers champion it.

**Missing References:**

Inter-coder agreement for computational linguistics -> offers a good read into the utility and shortcomings of IAA.

The paper above was updated and can be found part of a larger study into statistical methods for annotation analysis, part of this recent book: Statistical methods for annotation analysis

The aforementioned book starts by presenting coefficients of agreement, useful for assessing the reliability of an annotation scheme, then moves on to measures of dataset and annotator quality, to probabilistic models of annotation, in particular. On the latter topic, the following resources are also good starting points:

A recent EACL tutorial on 'Aggregating and Learning from Multiple Annotators' and these two TACL papers: The Benefits of a Model of Annotation, and Comparing Bayesian models of annotation

**Paper Topic And Main Contributions:**

The paper introduces a new inter annotator agreement metric,  HolisticIAA, based on sentence embedding similarity. The metric is compared against two traditional IAA metrics, Cohen's k and Krippendorff’s alpha, for the annotation process of one dataset on persuasion classification of text snippets.

**Questions For The Authors:**

I would like to see addressed the concerns raised under 'Reasons to reject'

**Reasons To Accept:**

The proposed metric allows to compute annotator agreement even if the annotators did not label the same documents, i.e., by pooling from the corpus similar sentences to those annotated by the annotators (using sentence embeddings) and computing label agreement on that set instead

**Reasons To Reject:**

Inter annotator agreement statistics are useful to measure the reliability of an annotation scheme, i.e, that the coders (aka annotators) have internalized the coding instructions s.t. a sufficient level of agreement can be observed), but are not informative about the quality of a dataset. Agreement is flawed for many reasons, e.g., agreement in mistakes, agreement due to label biases, large chance agreement in datasets with skewed classes, headline measurements with no information about the quality of the individual labels, and many more.

Also, I am not convinced about using as a form of evaluation the correlation with an existing IAA metric. Mainly because these metrics are already biased in some form, e.g., the kappa paradox.

Lastly, the quality of the sentence embeddings and the similarity thresholds seem central to the success of the proposed metric, however, their selection is treated rather lightly in the paper.

**Reproducibility:**

3: Could reproduce the results with some difficulty. The settings of parameters are underspecified or subjectively determined; the training/evaluation data are not widely available.

**Reviewer Confidence:**

4: Quite sure. I tried to check the important points carefully. It's unlikely, though conceivable, that I missed something that should affect my ratings.

**Typos Grammar Style And Presentation Improvements:**

line 537 -> Table* 6
line 646 -> without in advance, or early

---

> ### Author Rebuttal · Authors · 2023-08-29
>
> We thank the reviewer for the review and bringing references we were not aware of. Regarding the points raised under “Reasons to Reject”:
>
> “The usefulness of agreement as quality indication”:
>
> We reckon that IAA measures have several flaws, however despite these flaws they are still popular and mandatory to report when releasing new datasets. Our aim in this work rather than tackling all the flaws rightly pointed by the reviewer, is to propose one solution to tackle one specific aspect, i.e., the incomparability of annotations in different languages and different documents.
> Regarding the specific aspect of dataset quality, we are making the assumption that "coherent" annotation is an indicator of quality as it gives predictability. However, we could have made it more clear by reflecting it in the title and emphasizing on the concept of “coherence” instead of using the concept of “quality” (not the ideal in this context).
>
> “Bias of existing IAA metrics and impact on the evaluation”:
>
> The limitations of Cohen’s Kappa are indeed well known, nevertheless we needed to base our evaluation on existing and well established metrics and choose Cohen's Kappa as such basis. As such, until further proofs are made (i.e., studying the correlation with other IAA measures - envisaged as future work) one could consider for the moment our approach at worst as a “multi-lingual multi-document extension of Cohen’s Kappa”, which we believe would still be interesting per-se for the community in the context of organizing such large multi-lingual annotation campaigns.
>
> “Quality of the sentence embeddings and the similarity thresholds”:
>
> The primary goal of our submission was to share some lessons learned in a large-scale annotation campaign and to introduce the "concept" of a new way to compute IAA in a scenario where annotators do not label the same documents. Consequently, some approximations and simplifications were introduced, e.g., constraining the length of retrieved strings and their numbers, and the choice of the sentence embeddings model. However, we don’t claim the best set of parameters and models used, including the choice of the sentence embeddings model, but rather show that a versatile approach exists: for some applications or languages some embeddings could be more performant and the settings used in this paper should not be used, we could make clearer that these choices are task specific. We believe that in order to demonstrate the proof of concept of IAA agreement in the specifc scenario where annotators annotate different documents the choice of the underlying sentence embedding model is irrelevant. A follow-up study on comparison of different models could constitute a follow-up research direction
>
> Note on the dataset availability:  the curated dataset with persuasion technique annotations is available to the scientific community, and has been described in another paper, which is not disclosed here for anonymity reasons. The single-annotator versions thereof used to compute the various agreement metrics are envisaged to accompany the final version of this paper or to be made available upon request.

---

### Official Review · Reviewer_Wmuj · 2023-08-04

**Soundness:** 5

**Excitement:**

3: Ambivalent: It has merits (e.g., it reports state-of-the-art results, the idea is nice), but there are key weaknesses (e.g., it describes incremental work), and it can significantly benefit from another round of revision. However, I won't object to accepting it if my co-reviewers champion it.

**Missing References:**

Rebecca J. Passonneau and Bob Carpenter. 2014. The Benefits of a Model of Annotation. Transactions of the Association for Computational Linguistics, 2:311–326.

**Paper Topic And Main Contributions:**

This paper introduces a method called 'Holistic Inter-Annotator Agreement' to compute the (percentage) agreement between each pair of annotators on the semantically most similar sentences they have both annotated. This allows the authors to compute what annotators are most similar to a particular annotator. This ranking is shown to correlate well with the ranking obtained using standard IAA metrics such as Cohen's K.

**Questions For The Authors:**

1. Do I understand your objective correctly? Is your objective in effect to compute agreement between annotators on a larger scale, i.e., not only on the sentences they all annotate?

2. If the answer to the above is 'yes', how are you proposing to assess whether the obtained Holistic IAA value is sufficient to your purposes?

**Reasons To Accept:**

1. The paper is clearly built on a great deal of expertise about annotation, in particular in large-scale annotation projects. It raises some very good questions about the limitations of coefficients of agreement in such large-scale annotation projects, and illustrates some very good annotation practices - it could be useful to others aiming to run such a project.

2. The method itself makes a lot of sense.

3. the results obtained with the proposed method are analysed to a great depth.

**Reasons To Reject:**

1. The main problem I have with this paper is that it's not completely clear to me what is the problem that the authors are trying to solve. My understanding is that they are trying to come up with a more useful way of measuring the actual agreement between annotators not just on the few examples they both annotate, but I am not completely sure this is right.

2. A more general problem is that others have realized that coefficients of agreement are really a limited metric - especially in case of subjective judgments - and have tried to devise more informative approaches, although nobody as far as I know has addressed the specific issue tackled in this paper. I am especially thinking of the work on probabilistic models of annotation by Passonneau and Carpenter, and their TACL 2014 paper. In that paper, the authors argue convincingly that a single number cannot be sufficient  for the in-depth  analysis of annotation that the authors of this paper have in mind. The type of analysis they propose involve building models of each annotator, and of each item, that do allow for more insightful comparisons. I would encourage the authors to have a look at these methods and perhaps think about generalizing them.

**Reproducibility:**

4: Could mostly reproduce the results, but there may be some variation because of sample variance or minor variations in their interpretation of the protocol or method.

**Reviewer Confidence:**

4: Quite sure. I tried to check the important points carefully. It's unlikely, though conceivable, that I missed something that should affect my ratings.

---

> ### Author Rebuttal · Authors · 2023-08-29
>
> We thank the reviewer for the insightful review. Regarding the points raised and questions:
>
> Q1 “Objective of the reported work”:
> This work resulted from the specific need in our annotation campaign to capture agreement between annotators on a large scale, as standard IAA measures are limited in scope and utility in such a multi-lingual set-up. Hence, the answer to the direct question is indeed yes. It is important to emphasize that the proposed IAA metric is used to compute: (a)  agreement between annotators, and (b) final dataset coherency (the “dataset quality” might have been somewhat misleading wording we used in the manuscript). Instead of directly proving the appropriateness of the metric for the tasks at hand, we demonstrate the correlation of the new metric with well-established IAA metric in the community in the limited number of cases when such a comparison is possible, and we also demonstrate that after the second round of data curation the value of the new metric increases. We consider this as a qualitative demonstration of the metric to capture both IAA and coherence of the annotations.
>
> Q2 “Limitation of single-number agreement”:
> We agree on the limitation, and we were not familiar with the article mentioned, to which we will compare our approach. However, the approach proposed in our paper tries to emulate standard IAA measure with a single number as these are, despite their limitations, very popular due to their simplicity and even mandatory when publishing a dataset in order to give a rough idea of the quality of a dataset. We think having richer representation of agreement as described in the suggested paper is a very interesting direction for an extension of the work we report on in this paper.
> As regards subjectivity per se, we elaborate on this issue in the reply to reviewer 1, and we believe that (based on multiple rounds of meetings with the annotators) for the major fraction of the persuasion techniques the disagreements resulted mainly due to not sticking strictly to the definitions provided in the 60 pages of guidelines we prepared and bias resulting from professional background leading again to wrong comprehension of the persuasion technique definitions, e.g., simplifications have to follow a very strict logical pattern that needs to be identified in text, with which many people seemed to have problems. To conclude, we consider subjectivity as a minor factor contributing to the disagreements. We could add a section on these findings regarding subjectivity in the paper.

---

### Official Review · Reviewer_9ysf · 2023-08-05

**Typos Grammar Style And Presentation Improvements:** Please see lines
**Soundness:** 3

**Excitement:**

3: Ambivalent: It has merits (e.g., it reports state-of-the-art results, the idea is nice), but there are key weaknesses (e.g., it describes incremental work), and it can significantly benefit from another round of revision. However, I won't object to accepting it if my co-reviewers champion it.

**Missing References:**

-

**Paper Topic And Main Contributions:**

This paper is about the complexity of persuasion technique annotation in a large annotation campaign involving 6 languages and approximately 40 annotators. Its main contribution is introducing a new word embedding-based annotator agreement metric called HolisticIAA.

**Questions For The Authors:**

-

**Reasons To Accept:**

The annotation campaign done for their experiments is massive, and the article is well written.

**Reasons To Reject:**

Annotating persuasion techniques is a very subjective task. Moreover, the paper introduces the Holistic IAA metric but fails to explain how this is actually computed. Also, they conclude that Holistic IAA highly correlates with rankings computed using Cohen Kappa's in some settings, so it is not clear what the usefulness of this metric is.

**Reproducibility:**

3: Could reproduce the results with some difficulty. The settings of parameters are underspecified or subjectively determined; the training/evaluation data are not widely available.

**Reviewer Confidence:**

4: Quite sure. I tried to check the important points carefully. It's unlikely, though conceivable, that I missed something that should affect my ratings.

---

> ### Author Rebuttal · Authors · 2023-08-29
>
> We thank the reviewer for the fair review. Regarding the points raised under “Reasons to reject”:
>
> “Subjectivity of the annotation task”:
>
> In Section 4.3 we provide detailed analysis of the disagreements between different annotators vis-a-vis different persuasion techniques. While for some techniques (but few) it is indeed due to subjective perception, e.g. Loaded Language, Name Calling - due to cultural differences, we have found out that for the vast part of the techniques the high-level of disagreements (multiple meetings with annotators) was actually due to: (a) not understanding the techniques - many annotators did not read the guidelines thoroughly enough before starting the real annotations, which caused problems, for instance, as regards Simplifications, which have rather strict definitions and certain logic patterns have to be found in text (see the Annex of the paper); instead the annotators were often explicitly interpreting the word “simplification” and reasoning based on whether the presentation is too simplistic and whether certain facts were downplayed/exaggerated, which is actually a different technique, (b) professional bias - some of the media analysts who served as annotators were, unfortunately, often using background knowledge to make decision whether some text fragments are instances of persuasion techniques, which was strictly prohibited, (c) some of the annotators were making a direct link of persuasion technique labeling with claim checking and fact verification, which was again a mistake, etc. We have not provided in the paper a section on these findings due to space limitations although it could have been added in the final version of the paper to provide deeper insights on the lessons learned from the joint meetings with the annotators.
>
> "Computation of the holistic iaa value":
>
> The procedure to compute the value is described in section 5.1, however we reckon that there was no explicit formula for the computation in the submitted paper. We can easily provide such a formula:
>
> o(a1, a2, n_sample, th_len, th_sim) = mean([a1(x) == a(y) for x in A(a1) for y in Intersection( A(a2) , MostSimilar(x, n_sample) ) if len(x) <= th_len * len(y) and len(x) >= (1-th_len)*len(y) and sim(x, y, “LASER”) >= th_sim])  , where A(ai) is the set of annotated strings of annotator ai, and ai(x) is a function giving the annotation of annotator ai for the input string x.
>
> Please note, that this is an approximate value, as in the paper which considers only a subset of the annotations, and is asymmetric (the value is computed for a target annotator) using a maximal number of points to compare it with.
>
> We can provide also an exact formula with the parameters to compute an exact value which is symmetric for both annotators (not used in the paper for the moment):
>
> o(a1, a2, th_len, th_sim) = mean([a1(x) == a2(y) for x,y in A(a1) x A(a2) if len(x) <= th_len * len(y) and len(x) >= (1-th_len)*len(y) and sim(x, y, “LASER”) >= th_sim]).
>
> If introduced we will probably have the time to compare both approximate and exact values before the camera-ready deadline.
>
> “Usefulness of the approach”:
>
> The primary goal of our submission was to share some lessons learned in a large-scale annotation campaign and the concept of a new way to compute IAA in a scenario where annotators do not label the same documents and to compare annotations across different languages to guarantee certain level of coherence. While the introduced IAA metric is just a proof of concept, and certain  approximations and simplifications were introduced/made, e.g., constraining the length of retrieved strings and their numbers, and the choice of a specific sentence embeddings model, it helped to circumvent the limited scope and utility of IAA in such a large-scale multilingual campaign. Disregarding the fact of some limitations of the results reported in our manuscript, we believe the reported experiments might have practical value for scientists involved in suchlike large annotation campaigns, and the presented work might trigger further studies in this direction.
>
> Note on the dataset Dataset availability:  the  curated dataset with persuasion technique annotations is available to the scientific community, and has been described in another paper, which is not disclosed here for anonymity reasons. The single-annotator versions thereof used to compute the various agreement metrics are envisaged to accompany the final version of this paper or to be made available upon request.

---

### Meta-Review · Area_Chair_4mgs · 2023-09-15

**Recommendation:** 4

**Metareview:**

The paper discusses a new idea for assessing IAA and annotation quality in large multilingual annotation campaigns characterized by high complexity and minimal overlap in the annotated items. Specifically, the authors introduce a new annotator agreement metric based on word embeddings to better account for the difficulties inherent in human annotation of persuasion techniques and to assess the overall coherence of a dataset. Given the subjective nature of the task, investigations into annotators’ disagreement on the specific techniques can significantly help improve future research. The rebuttal phase was constructive and helped clarify several points. I am confident that the authors will enhance the revised version by incorporating many of the details mentioned in their responses.

**Pros**
- it introduces a novel, robust method for assessing agreement in large multilingual annotation campaigns, where overlap between annotators is limited;
- it presents a new metric as a proof of concept and adequately highlights the limitations of the approach;
- a large-scale annotation campaign was conducted (6 languages and about 40 annotators);
- the paper reveals high expertise ;
- the work can be of considerable value for other researchers in the field;
- the article is well written and clear;
- results analysis is sufficiently thorough.

**Cons**
- greater clarity seems to be required in stating and explaining the main objectives of the paper, to avoid misunderstanding. The responses provided by the authors during the clarification phase should be integrated into the revised version;
- a discussion on the role of subjectivity and on the main sources of disagreement should be integrated;
- the explicit formula for the new metric needs to be given in the paper;
- essential details about the experimental setting are missing (see reviewers comments).
It is worth noting that these deficiencies have been convincingly addressed in the author responses, suggesting that they would necessitate only minor revisions in the final paper.

---

### Decision · Program_Chairs · 2023-10-07

**Decision:**

Accept-Main

**Comment:**

The paper discusses a new idea for assessing IAA and annotation quality in large multilingual annotation campaigns characterized by high complexity and minimal overlap in the annotated items. Specifically, the authors introduce a new annotator agreement metric based on word embeddings to better account for the difficulties inherent in human annotation of persuasion techniques and to assess the overall coherence of a dataset. Given the subjective nature of the task, investigations into annotators’ disagreement on the specific techniques can significantly help improve future research. The rebuttal phase was constructive and helped clarify several points. I am confident that the authors will enhance the revised version by incorporating many of the details mentioned in their responses.

**Pros**
- it introduces a novel, robust method for assessing agreement in large multilingual annotation campaigns, where overlap between annotators is limited;
- it presents a new metric as a proof of concept and adequately highlights the limitations of the approach;
- a large-scale annotation campaign was conducted (6 languages and about 40 annotators);
- the paper reveals high expertise ;
- the work can be of considerable value for other researchers in the field;
- the article is well written and clear;
- results analysis is sufficiently thorough.

**Cons**
- greater clarity seems to be required in stating and explaining the main objectives of the paper, to avoid misunderstanding. The responses provided by the authors during the clarification phase should be integrated into the revised version;
- a discussion on the role of subjectivity and on the main sources of disagreement should be integrated;
- the explicit formula for the new metric needs to be given in the paper;
- essential details about the experimental setting are missing (see reviewers comments).
It is worth noting that these deficiencies have been convincingly addressed in the author responses, suggesting that they would necessitate only minor revisions in the final paper.